# Compliance with a Procalcitonin-Based Protocol in Patients with Ventilation-Associated Pneumonia: An Observational, Retrospective Study

**DOI:** 10.3390/antibiotics12071208

**Published:** 2023-07-20

**Authors:** Matthieu Rossi, Louis Delamarre, Gary Duclos, Ines Lakbar, Emmanuelle Hammad, Charlotte Arbelot, Laurent Zieleskiewicz, Marc Leone

**Affiliations:** Assistance Publique Hôpitaux de Marseille, Department of Anesthesiology and Intensive Care, Hôpital Nord, Aix-Marseille University, 13015 Marseille, France; louis.delamarre@ap-hm.fr (L.D.); gary.duclos@ap-hm.fr (G.D.); ines.lakbar@ap-hm.fr (I.L.); emmanuelle.hammad@ap-hm.fr (E.H.); charlotte.arbelot@ap-hm.fr (C.A.); laurent.zieleskiewicz@ap-hm.fr (L.Z.); marc.leone@ap-hm.fr (M.L.)

**Keywords:** compliance, procalcitonin, pneumonia, intensive care

## Abstract

Background: Procalcitonin (PCT) protocols to guide antibiotic treatment for ventilator-associated pneumonia (VAP) in the intensive care unit aim at reducing antibiotic exposure. Our study goal was to measure compliance with a PCT protocol for VAP and to determine the associated variables. Methods: From 2017 to 2021, we conducted a retrospective, monocentric study including patients treated for VAP. In our PCT protocol, PCT was measured at the initiation of antibiotic treatment and every 48 h until treatment completion; antibiotics were stopped if PCT decreased by more than 80% from its highest value or fell below 0.5 ng/mL. We assessed the compliance with the PCT protocol and compared the compliant and noncompliant groups. Results: Among the 177 included patients, compliance with the PCT protocol was assessed at 58%. Noncompliance was due to lack of PCT measurements in 76% of cases. Compliance was higher in the medical patients (*p* = 0.04) and in those admitted for SARS-CoV-2 (*p* = 0.02). Compliance regarding the interruption of antibiotic therapy based on PCT was lower on weekends and holidays (*p* = 0.01). Outcomes did not differ according to compliance. Conclusion: This study assessed real-life compliance with the PCT protocol to monitor antibiotic treatment for VAP. Improving the measurement of PCT at the bedside would increase the rate.

## 1. Introduction

Antibiotic treatment of ventilator-associated pneumonia (VAP) is a daily challenge in the intensive care unit (ICU) [1]. There is currently a trend toward a shorter antibiotic course with a recommended treatment duration of 7 days [2,3]. However, ICU patients often receive antibiotics for longer than recommended with the risks of emergence of antimicrobial resistance and complications related to prolonged administration [2].

Procalcitonin (PCT) has been evaluated in the setting of bacterial infection for guiding clinicians to initiate and monitor antibiotic treatment. During sepsis, PCT becomes ubiquitous through the activation of proinflammatory cytokines; serum concentrations are therefore rapidly increased [4]. However, in the ICU, PCT is not reliable for ruling out the bacterial nature of an infection [5,6]. Monitoring serum PCT concentrations may help the clinician to decide the duration of antibiotic treatment. Several randomized controlled trials and one systematic review showed that monitoring serum PCT reduced antibiotic exposure [7,8,9,10], without impairing survival [7]. Consequently, serum PCT-guided antibiotic duration is suggested by several guidelines [2,3].

In the literature, only few articles report compliance with ICU protocols [11,12]. To our knowledge, compliance with a PCT protocol to determine antibiotic duration for patients with confirmed VAP has never been used in this clinical setting. Here, we propose to measure real-life compliance with a serum PCT-guided antibiotic protocol for confirmed VAP in a single ICU. We hypothesized that compliance with protocol depends on patient-, clinician-, and time-related factors.

The first objective of our study was to describe the compliance rate with a protocol for monitoring the antibiotic duration using serum PCT in patients with confirmed VAP. The secondary objectives were to determine the variables associated with compliance, and the association between weekends/holidays and compliance. We also assessed the association between protocol compliance and the following variables: ICU and 28-day mortality, duration of ICU stay, duration of mechanical ventilation, and antibiotic exposure.

## 2. Results

### 2.1. Patient Characteristics

During the study period, 3303 patients were admitted to the ICU, of whom 355 met the eligibility criteria. Among them, 177 patients were included in the final analysis (Figure 1). The mean age of patients was 59 ± 17 years; women accounted for 25% of patients, and the SAPS II score at the ICU admission was 50 (IQR 22). Medical patients and severe trauma patients represented 43.5% and 36.2% of our cohort, respectively. One hundred and nineteen (67%) patients and 57 (32%) patients had criteria for ARDS and septic shock, respectively. Risk factors for multidrug-resistant pathogen carriage were found in 37 (21%) patients, and 37 (21%) received at least one antibiotic in the previous 15 days. Eighty (45%) patients were treated by selective digestive decontamination (Table 1).

### 2.2. Main Objective

Compliance was found in 103 (58%) patients. Early interruption of antibiotic treatment according to the PCT protocol was achieved in 36 (35%) patients. The noncompliant group included 74 (42%) patients. A lack of PCT measurement was found in 56 (76%) patients, while antibiotic treatment was continued in disagreement with the PCT protocol in 16 (21%) patients (Table 2).

### 2.3. Secondary Objectives

Gender, age, comorbidities, severity of disease (septic shock and ARDS), type of bacteria, and multidrug-resistant pathogen carriage were not associated with compliance. In contrast, compliance was 68% in the medical patients versus 51% in the nonmedical patients (*p* = 0.04). Compliance was also higher in the patients with SARS-CoV-2 pneumonia (32 (74%) vs. 71 (53%) in COVID-19-positive vs. COVID-19-negative patients, respectively, *p* = 0.021). Of note, higher rates of noncompliance in stopping antibiotic therapy based on PCT was observed during weekends and holidays (82.2% on weekends and holidays vs. 57.8% otherwise, *p =* 0.011).

The assessment of compliance over time is shown in Figure 2a. No significant time effect was reported (*p* = 0.11). However, the complete achievement of the repeated PCT measurements according to the PCT protocol increased significantly during the study period (*p* = 0.012) (Figure 2b).

The ICU mortality and day 28 mortality rates in the ICU did not differ between the compliant and noncompliant groups (*p* = 0.74 and *p* = 0.87, respectively). The duration of the ICU stay (*p* = 0.73), the duration of invasive mechanical ventilation (*p* = 0.85), and the duration of antibiotic treatment did not differ in the two groups (7.4 ± 4.4 days versus 7.5 ± 3.1 days, *p* = 0.09) (Table 3).

## 3. Discussion

In our cohort, compliance with the PCT protocol was achieved in 58% of patients. We observed that compliance was higher among medical and SARS-CoV-2 patients. However, compliance in the interruption of antibiotic therapy based on PCT was lower during weekends and holidays. There was a slight trend toward improved protocol compliance over the years. Nevertheless, compliance with the PCT protocol did not have any significant effects on outcomes, particularly the duration of antibiotic treatment.

In our patients, compliance with the PCT protocol was assessed at 58%. Previous studies assessing the adherence to a PCT protocol for both initiating and interrupting antibiotic treatment showed rates from 47.7% to 51.6% [13,14], which is below that obtained in our study. The difference between the two studies could be attributed to a time effect, as our study was conducted later. Another hypothesis is related to ICU organization. Indeed, our protocols were written, accessible on our website, and weekly infectious case review meetings probably contributed to the improvement in compliance. Even if rates below 60% are not optimal, one should keep in mind that a multicenter observational study found an average compliance rate of 26% among 660 ICU patients [12].

The primary reason for noncompliance was the absence of serum PCT measurements in 75% of cases. When blood samples were collected for serum PCT measurement, the antibiotic treatment was in agreement with the PCT protocol in 85% of cases. This underlines the importance of measuring serum PCT concentrations, especially during the on-call periods. Another way for improvement is the automation of biological prescription, as we implemented during the COVID-19 pandemic, which resulted in improved compliance among those patients.

Among 24.3% of patients with no compliance with the PCT protocol despite measurements of serum PCT concentrations, the clinical picture probably supported the medical decision. However, we cannot exclude that the PCT results were not seen at the time by the medical team. As the cost of PCT dosages is significant, this issue should be resolved in the future. The decreased compliance during weekends and holidays, regarding the interruption of antibiotic therapy based on PCT, which corresponded to days with a small team on-site, seems in line with this last hypothesis. In the literature, the effects of on-call periods are variable [15,16]. One should note that in our ICU, a senior intensivist was on-call on-site 24 h, while a second senior intensivist, who was on-call off-site, participated to around day 7.

Importantly, our PCT protocol did not impact patient outcomes. Measuring serum PCT concentrations has been shown to reduce antibiotic exposure, as demonstrated by large multicenter randomized controlled trials [7,9], and this result was associated with decreased costs [17,18]. Thus, the lack of effect of the PCT protocol on antibiotic exposure raises questions about its relevance, although it is in agreement with a Cochrane systematic review [19]. Our findings differ from a previous study that reported a significant reduction in the treatment duration in the PCT group [14]. However, the durations of treatment of our controls were shorter than those of this previous study.

Our study has several limitations. Firstly, we selected only patients with a definitive diagnosis of VAP. Thus, we excluded those with suspected VAP in whom the antibiotics were interrupted early due to a diagnosis refinement. In these patients, the use of PCT could participate to the decision to prematurely interrupt antibiotics. This could explain the lack of difference in terms of antibiotic exposure in our study. Secondly, the retrospective nature itself is a limitation in the interpretation of observational studies. Thirdly, our collection of data did not include the evolution of patients in terms of severity; one can suggest that compliance decreased if serum PCT concentrations were low despite an increase in severity (and vice versa). Lastly, the experience of senior physicians may have been included in our analysis. However, as most of our decisions are collective, this would not reflect our real-life practices.

## 4. Materials and Methods

Our study was an observational, monocentric, retrospective study conducted between 20 December 2017 and 30 April 2021 in the polyvalent ICU (15 beds) of the Hospital North within the Assistance Publique Hôpitaux Universitaires de Marseille, Marseille, France.

### 4.1. Ethics Statement

This retrospective, monocentric, noninterventional study was based on the review of clinical and laboratory medical records. Under French law, patient consent was not required for this type of noninterventional study, provided the patients had received information about the potential use of anonymized medical data for research purposes and retained the right to oppose it (ethics committee approval by SFAR CERAR committee: IRB-00010254-2022-001; CNIL authorization: PADS21_176) [20]. The STROBE checklist for observational studies was used to report the results of our research [21].

### 4.2. Patient Management

Patients with VAP were treated according to guidelines [2,3,22,23]. Patients were assessed daily to confirm ongoing suspicion of disease; antibiotic treatments were re-assessed daily, and the anti-infective agents were discussed once a week with our infectious disease specialists. Clinicians had to consider daily the possibility to stop treatment in the event of negative cultures. Selective digestive decontamination, as described previously [24], was used for all patients admitted to our ICU, with the exception of those transferred from wards or from the operating room for surgical complications.

### 4.3. Data Collection

The list of patients was collected from the diagnosis coding service (ICD-10 nomenclature) and medical procedures (French CCAM nomenclature) of our institution. The codes used for the research were, on the one hand, the codes corresponding to bacterial pneumonia and, on the other hand, the codes for mechanical ventilation procedures. Retained patients were those who had both a bacterial pneumonia code and a mechanical ventilation code. Ventilator-associated pneumonia was defined according to international definition [23]. If a patient had several episodes of VAP during the ICU stay, only the first episode was considered in our analysis.

### 4.4. Inclusion and Exclusion Criteria

During the study period, eligible patients included all patients admitted to our ICU who were both coded for bacterial pneumonia and mechanical ventilation. The study period started with implementation of PCT protocol in March 2017. Patients under 18 years of age, those with coding errors, those with ICU stays below three days, the second episode of VAP, and pregnant women were excluded from the analysis.

### 4.5. Data Collected

All study data were extracted retrospectively from the institutional computerized patient record. Data collected included selected demographic data, such as age, gender, simplified acute physiology score (SAPSII) at admission, and mortality rates in the ICU and at 28 days. We also collected:at ICU admission: reason for admission, prior medical history, risk factors for multidrug-resistant pathogen carriage, and existence of antibiotic treatment in the previous three months;during the ICU stay: duration of mechanical ventilation, identified germ(s), antibiotic(s) used, date of the onset of antibiotic treatment, duration of antibiotic treatment, and failure of the first line of antibiotic treatment;at the onset of antibiotic treatment: presence of acute respiratory distress syndrome (ARDS), presence of septic shock, plasma creatinine and the use of extrarenal replacement therapy, and first day of antibiotics (on-call days (i.e., weekends and holidays) versus working hours);during the antibiotic treatment for VAP: serum PCT concentrations.

Our PCT protocol derived from the SAPS trial: all patients receiving antibiotics for a suspected infection had a serum PCT measurement on the day of initiation of antibiotic treatment and then every 48 h [7]. The first day of serum PCT measurement was defined as the first day of antibiotic treatment, independent of the microbiological documentation. Serum PCT concentrations were recorded on the initial day and then every 48 h. If serum PCT decreased by at least 80% from its highest value or if it fell below or equal to 0.5 ng/mL, the antibiotic treatment was stopped. In the other cases, antibiotics were continued until the date originally planned and according to the standard of care for the duration of treatment, i.e., 7 days according to guidelines [2,3,22,23]. Initiation of antibiotic treatment did not rely on serum PCT concentrations.

Patients were classified into two groups based on the compliance with PCT protocol. Compliance was defined by:Serum PCT measurements were performed as required by the PCT protocol;Antibiotic treatment was adapted to serum PCT concentrations in agreement with the PCT protocol.
Two groups were identified:
Compliant group: antibiotic treatment was continued because serum PCT concentrations did not decrease under the protocol threshold, and antibiotic treatment was interrupted because serum PCT concentrations decreased according to PCT protocol;Noncompliant group: serum PCT measurements were not performed, or antibiotic treatment did not respect the PCT protocol. Four cases were identified: (1) PCT not available during the antibiotic treatment; (2) PCT partially available during the antibiotic treatment; (3) continuation of antibiotics despite PCT protocol indicating its interruption, and (4) interruption of antibiotic treatment despite PCT protocol indicating its continuation.

The human factor was assessed by determining whether the day on which the protocol was not respected corresponded to the on-call periods defined by weekends and holidays.

### 4.6. Statistical Analysis

Due to the exploratory nature of the study, no specific sample size was defined. Data are expressed as numbers and percentages for categorical data and as mean and standard deviations (or median and 1st–3rd percentiles, when appropriate) for quantitative variables. Missing data were omitted from analyses without imputation because of the expected small sample size. Outliers were searched for and removed, without replacement.

Comparisons of categorical variables were performed using a Fisher’s exact test when 2 groups were considered and by a multiclass Chi2 test (χ2) in other cases. Comparisons of quantitative variables between groups were performed using a nonparametric Mann–Whitney test (if two groups were considered) or a Kruskal–Wallis test (if more than two groups were considered).

A *p*-value < 0.05 was required to reach statistical significance. The statistical tests were bilateral. Analysis was performed using R 4.0.4 software (R Core Team, R Foundation for Statistical Computing, Vienna, Austria, 2021) [25].

## 5. Conclusions

Our study shows that real-life compliance with the PCT protocol aiming at monitoring the duration of antibiotic treatment was 58%. Noncompliance was mainly explained by a lack of PCT measurements. Compliance with the PCT protocol was increased for medical patients including those admitted for SARS-CoV-2 pneumonia and was reduced on weekends and holidays, regarding the interruption of antibiotic therapy based on PCT. No association was found between compliance with the PCT protocol and outcomes, questioning the relevance of serum PCT measurements.

## Figures and Tables

**Figure 1 antibiotics-12-01208-f001:**
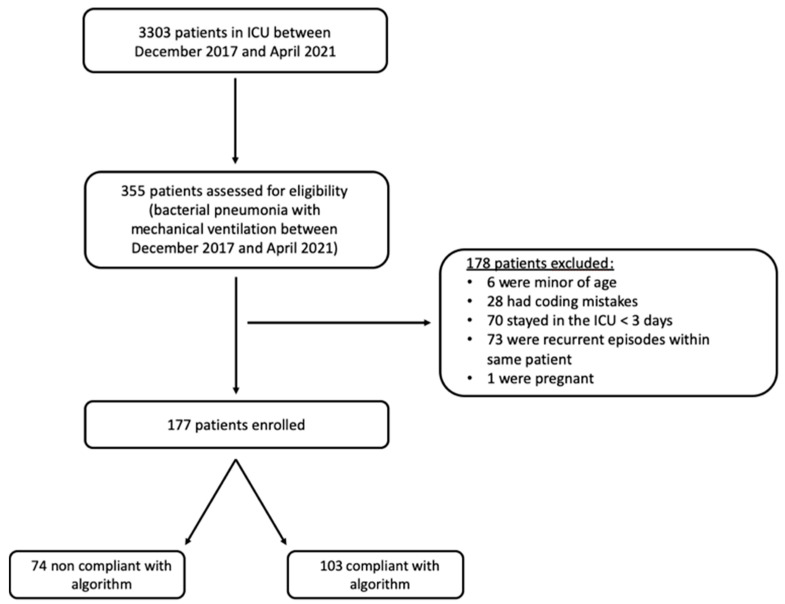
Flow chart of the study.

**Figure 2 antibiotics-12-01208-f002:**
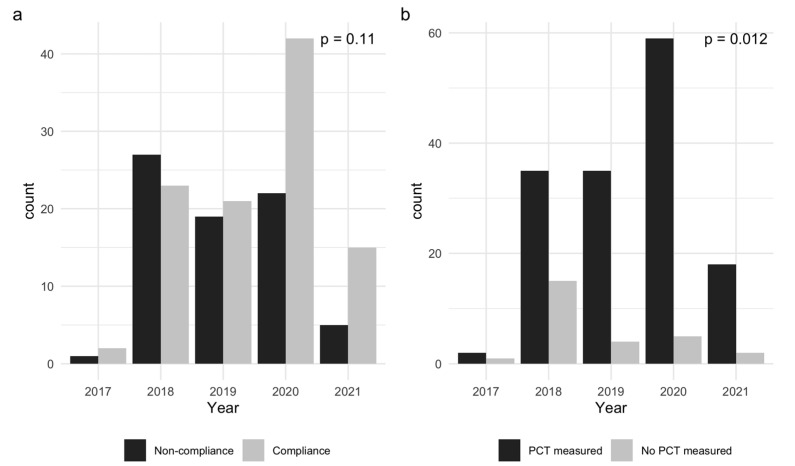
(**a**) Assessment of compliance over time; (**b**) assessment of complete achievement of PCT measures according to protocol over time.

**Table 1 antibiotics-12-01208-t001:** Features of patients according to compliance with procalcitonin protocol.

	Total	Noncompliance	Compliance	*p*
	(n = 177)	(n = 74)	(n = 103)	
Gender (female)	44 (24.9%)	16 (36.4%)	28 (63.6%)	0.504
Age, average (SD), years	58.7 (16.5)	57.8 (17.0)	59.3 (16.2)	0.549
**Reason for admission**				0.13
Elective surgery	22 (12.4%)	10 (45.5%)	12 (54.5%)	
Medical	77 (43.5%)	25 (32.4%)	52 (67.6%)	
Trauma	64 (36.2%)	30 (46.9%)	34 (53.1%)	
Urgent surgery	13 (7.3%)	8 (61.5%)	5 (38.5%)	
SARS-CoV-2 pneumonia	43 (24.3%)	11 (25.6%)	32 (74.4%)	
BMI, mean (SD)	26.0 (6.2)	25.9 (5.6)	26.1 (6.6)	0.881
Hypertension	73 (41.2%)	30 (41.1%)	43 (58.9%)	0.995
Obesity	35 (19.8%)	15 (42.9%)	20 (57.1%)	1
Diabetes	36 (20.3%)	16 (44.4%)	20 (55.6%)	0.865
Coronary heart disease	23 (13.0%)	9 (39.1%)	14 (60.9%)	0.958
Chronic heart failure	7 (4.0%)	2 (28.6%)	5 (71.4%)	0.739
Chronic kidney disease	6 (3.4%)	4 (66.7%)	2 (33.3%)	0.404
COPD	21 (11.9%)	9 (42.9%)	12 (57.1%)	1
Obstructive sleep apnea	9 (5.1%)	1 (11.1%)	8 (88.9%)	0.117
Chronic respiratory failure	7 (4.0%)	3 (42.9%)	4 (57.1%)	1
Cirrhosis	2 (1.1%)	1 (50%)	1 (50%)	1
**Risk factors for MDR infection**				
Institutionalized	0 (0%)	0 (0%)	0 (0%)	1
Hospitalization in the last 3 months	31 (17.5%)	13 (42.0%)	18 (58.0%)	1
Antibiotic therapy in the last 3 months	18 (10.2%)	5 (27.8%)	13 (72.2%)	0.307
History of MDR infection	6 (3.4%)	2 (33.3%)	4 (66.7%)	0.994
Recent endemic-country travel	4 (2.3%)	0 (0%)	4 (100%)	0.229
Antibiotic therapy within last 15 days	37 (20.9%)	11 (29.8%)	26 (70.2%)	0.137
Selective digestive decontamination	80 (45.2%)	38 (47.5%)	42 (52.5%)	0.215
SAPS II admission, mean (SD)	49.9 (15.1)	49.6 (13.8)	50.1 (16.0)	0.828
Intubation at admission	171 (96.6%)	70 (41.0%)	101 (59.0%)	0.404
Ventilator-associated pneumonia	156 (88.1%)	65 (41.7%)	91 (58.3%)	1
Pneumonia requiring mechanical ventilation	21 (11.9%)	9 (42.9%)	12 (57.1%)	1
Surgical site infection	13 (7.3%)	4 (30.8%)	9 (69.2%)	0.585
Septic shock	57 (32.2%)	27 (47.4%)	30 (52.6%)	0.384
Maximum norepinephrine, median [IQR], mg/h	3.00 [2.3; 3.5]	3.00 [2.3; 3.9]	3.00 [2.3; 3.3]	0.576
ARDS	119 (67%)	48 (40.3%)	71 (59.7%)	0.62
Mechanical ventilation time, median [IQR], days	13.0 [8; 25]	12.0 [7; 25]	13.0 [8; 25]	0.5
Duration of antibiotic treatment, mean (SD), days	7.43 (3.9)	7.46 (3.1)	7.41 (4.4)	0.928
Renal replacement therapy	5 (2.8%)	2 (40%)	3 (60%)	1

n: size; %: percentage; SD: standard deviation; BMI: body mass index; COPD: chronic obstructive pulmonary disease; IQR: interquartile range; MDR: multidrug resistance; SAPS II: Simplified acute physiology score (2nd version); ARDS: acute respiratory distress syndrome.

**Table 2 antibiotics-12-01208-t002:** Reasons for classification in the compliant or noncompliant group.

	Noncompliance	Compliance
	(n = 74) (%)	(n = 103) (%)
**Lack of measurement of PCT**	**56 (75.6%)**	
Incomplete PCT measuring scheme	29 (39.1%)	
PCT never measured	27 (36.4%)	
**Noncompliance with the protocol despite dosage**	**18 (24.3%)**	
Antibiotic discontinuation against PCT	2 (2.7%)	
Antibiotic continuation against PCT	16 (21.6%)	
Antibiotic discontinuation according to PCT		36 (35%)
Antibiotic continuation according to PCT		63 (61.1%)
Noninformative PCT (<0.1 ng/mL)		4 (3.9%)

PCT: Procalcitonin.

**Table 3 antibiotics-12-01208-t003:** Outcome according to procalcitonin protocol compliance.

	Total Cohort	Noncompliance	Compliance	*p*
	(n = 177)	(n = 74)	(n = 103)	
**Death due to pneumonia**				
No	164 (92.7%)	69 (93.2%)	95 (92.2%)	1
Yes	13 (7.3%)	5 (6.8%)	8 (7.8%)	
**Death in intensive care**				
No	124 (70.1%)	53 (71.6%)	71 (68.9%)	0.739
Yes	53 (29.9%)	21 (28.4%)	32 (31.1%)	
**Mortality at 28 days**				
No	121 (68.4%)	50 (67.6%)	71 (68.9%)	0.873
Yes	56 (31.6%)	24 (32.4%)	32 (31.1%)	
**Length of stay in intensive care**				
Mean (SD), days	25.0 (19.9)	25.7 (23.7)	24.5 (16.8)	0.731
Median [IQR], days	20.0 [11.0–34.0]	19.0 [11.0–34.0]	20.0 [12.0–34.5]	
**Mechanical ventilation time**				
Mean (SD), days	18.6 (18.3)	19.8 (23.1)	17.7 (13.9)	0.854
Median [IQR], days	13.0 [8.00–25.0]	12.0 [7.00–25.0]	13.0 [8.00–25.0]	
**Time of antibiotic exposure**				
Mean (SD), days	7.43 (3.92)	7.46 (3.14)	7.41 (4.42)	0.088
Median [IQR], days	7.00 [5.00–7.00]	7.00 [7.00–7.00]	7.00 [5.00–7.00]	
**Success of the first antibiotic therapy**				
No	34 (19.2%)	12 (16.2%)	22 (21.4%)	0.34
Yes	142 (80.2%)	61 (82.4%)	81 (78.6%)	

SD: standard deviation; IQR: interquartile range; %: percentage.

## Data Availability

Restrictions apply to the availability of these data. Data were obtained from the Assistance Publique des Hôpitaux de Marseille and are available from the authors with the permission of the Assistance Publique des Hôpitaux de Marseille.

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
