# Peer review of "Compliance with a Procalcitonin-Based Protocol in Patients with Ventilation-Associated Pneumonia: An Observational, Retrospective Study"

_antibiotics, 2023, doi:10.3390/antibiotics12071208_

Round 1

Reviewer 1 Report

Thank you for submitting this manuscript. This research is interesting. I have few points to be addressed:

1- The "conclusion" part in the abstract should be re-written. no need for numbers or result duplication to be in this part of the abstract.

2- Suggestion: the percentages (%) in table 1 are to be calculated for each subgroup. Example:

Elective surgery 22 (12.4%) non-compliance: 10 (13.5%) compliance: 12 (11.7%)

to be written as 22 (12.4%) non-compliance: 10 (45.5%)  compliance: 12 (54.5%)

3- Table 4: Don't use 0 & 1 coding

4- Discussion: needs English language revision.

The discussion part needs English language revision

Author Response

Reviewer 1

Dear reviewer,

We thank you for your insightful suggestions and comment. We have answered your questions, as detailed below, and amended the manuscript based on your suggestions.

We hope those changes will increase the readability and the pertinence of our manuscript.

Kind regards,

Matthieu Rossi

Question 1: The "conclusion" part in the abstract should be re-written. no need for numbers or result duplication to be in this part of the abstract.

Answer: We thank you for this suggestion. We have modified it to make it clearer and less redundant, as follows: This study assessed real-life compliance with PCT-Protocol to monitor antibiotic treatment for VAP. Improving the measurement of PCT at the bedside would increase the rate.”

Question 2:  Suggestion: the percentages (%) in table 1 are to be calculated for each subgroup. Example:

Elective surgery 22 (12.4%) non-compliance: 10 (13.5%) compliance: 12 (11.7%)

to be written as 22 (12.4%) non-compliance: 10 (45.5%) compliance: 12 (54.5%)

Answer: We thank Reviewer 1 for spotting these misleading elements. We have modified table 1 as suggested, to make those within-group percentages clearer.

Question 3: Table 4: Don't use 0 & 1 coding     

Answer: We thank you for pointing this misleading coding. We have modified the “0/1” coding in favor of a “No/Yes” coding.

Question 4: Discussion: needs English language revision

            Answer: We thank you for this pertinent suggestion. Professional English editing has been performed to improve the clarity and readability of this manuscript. We hope it will meet your expectations.

Reviewer 2 Report

Introduction

Line 52: “We also assessed the association between the ICU and 28 days mortality rates, the 52 durations of ICU stay, mechanical ventilation, and antibiotic exposure.” It is not clear what is the relationship between this assessment and the outcome presented or the relationship between this information and the compliance. Please define better or retire from here if it is only descritive. It seems to respond to a different quetion: “Does compliance affect the results of mortality and length of stay among patients with PCR protocol?”

Results

Table 1

Maximum norepineprhin results, present only the estimator according to the distribution of the data (mean for normal, median for others).

The same recommendation for the mechanical ventilation.

Table 3. According to the title: “Variables associated with procalcitonin protocol compliance. ”

However, it is only a bivariate relationship with the outcome (compliance), no specific data is shown, only a p value, that is not informative.

Methods:

Line 220 – “The STROBE checklist 220 for observational studies was used to report the results of our research “. It doesn´t seem to comply with strobe. No indication of the type of study was found

Line 296. The statistical analysis does not allow to answer “The secondary objectives were to determine the variables associated with compliance, the 50 effect of year on compliance and the association between weekends/holydays and compliance”. It does not allow to understand if there is a real relationship between those factors observed and thefinal compliance. To be able to be published I think it would need a multivariate analysis. I don´t think it would be publish without such analysis since it might allow to have a better understanding of the problem explored.

Author Response

Reviewer 2

Dear reviewer,

We thank you for your reviewing. Here are the items we modified thanks to your work.

Kind regards,

Matthieu Rossi

Question 1:

Introduction
Line 52: “We also assessed the association between the ICU and 28 days mortality rates, the durations of ICU stay, mechanical ventilation, and antibiotic exposure.” It is not clear what is the relationship between this assessment and the outcome presented or the relationship between this information and the compliance. Please define better or retire from here if it is only descritive. It seems to respond to a different quetion: “Does compliance affect the results of mortality and length of stay among patients with PCR protocol?”

Answer: We thank you for this insightful question. Indeed, we aimed to measure the association between protocol compliance and the following variables: mortality, length of ICU stay, length of mechanical ventilation and antibiotic exposure. We corrected this misleading wording as follows.

From: “We also assessed the association between the ICU and 28 days mortality rates, the durations of ICU stay, mechanical ventilation, and antibiotic exposure.” To: “We also assessed the association between protocol compliance and the following variables : ICU and 28-day mortality, duration of ICU stay, duration of mechanical ventilation, and antibiotic exposure.”

Question 2: Results > Table 1

Maximum norepineprhin results, present only the estimator according to the distribution of the data (mean for normal, median for others).

The same recommendation for the mechanical ventilation.

Answer: We thank you for pointing this redundancy. We modified the table 1 to display only the median, due to non-normal data distribution. We have modified the Norepinephrine and Mechanical ventilation rows as follows:

            From:                         

Maximum norepinephrine, Mean (SD), mg/hour

3.18 (1.67)

3.32 (1.80)

3.06 (1.57)

0.576

Maximum norepinephrine, Median [IQR], mg/hour

3.00 [2.3;3.5]

3.00 [2.3; 3.9]

3.00 [2.3; 3.3]

Mechanical ventilation time, Mean (SD), days

18.6 (18.3)

19.8 (23.1)

17.7 (13.9)

0.5

Mechanical ventilation time, Median [IQR], days

13.0 [8; 25]

12.0 [7; 25]

13.0 [8; 25]

            To:

Maximum norepinephrine, Median [IQR], mg/hour

3.00 [2.3;3.5]

3.00 [2.3; 3.9]

3.00 [2.3; 3.3]

0.576

Mechanical ventilation time, Median [IQR], days

13.0 [8; 25]

12.0 [7; 25]

13.0 [8; 25]

0.5

Question 3: Table 3. According to the title: “Variables associated with procalcitonin protocol compliance.

However, it is only a bivariate relationship with the outcome (compliance), no specific data is shown, only a p-value, that is not informative.

Answer: We thank you for pointing this lack of clarity. This table 3 is largely redundant with the table 1, in that it displays a selected set of variables to present the significance of their association with compliance. To improve readability, we decided to delete this table 3 and report the significant associations in the text directly. Please note that we refined the definition of the variable “Non-compliance with protocol during weekend days and holidays” as “non-compliance in stopping antibiotic therapy based on procalcitonin during a weekend day or holiday”, as the only non-compliant action that can be narrowed down to a specific day is this item.

We thus modified the paragraph 2.3. Secondary objectives as follows:

From: Gender, age, comorbidities, severity of disease (septic shock, ARDS), type of bacteria and multidrug-resistant pathogen carriage were not associated with compliance (Table 3). In contrast, compliance was 68% in the medical patients versus 51% in the non-medical patients (p = 0.04). Of note, compliance was also higher in the patients with SARS-CoV-2 pneumonia (32 (74%) vs. 71(53%) in COVID-19 positive vs. COVID-19 negative patients, respectively, p = 0.021). […]. On-call days were associated with decreased compliance (23% vs. 50%, p = 0.011) (Table 3).“

To: “Gender, age, comorbidities, severity of disease (septic shock, ARDS), type of bacteria and multidrug-resistant pathogen carriage were not associated with compliance (Table 3). In contrast, compliance was 68% in the medical patients versus 51% in the non-medical patients (p = 0.04). Compliance was also higher in the patients with SARS-CoV-2 pneumonia (32 (74%) vs. 71(53%) in COVID-19 positive vs. COVID-19 negative patients, respectively, p = 0.021). Of note, higher rates of non-compliance in stopping antibiotic therapy based on procalcitonin was observed during weekend days or holidays (82.2% in weekend days or holidays vs. 57.8% otherwise, p = 0.011). […]”

Question 4: Methods:

Line 220 – “The STROBE checklist 220 for observational studies was used to report the results of our research “. It doesn´t seem to comply with strobe. No indication of the type of study was found

Answer : We thank you for your comment. Nevertheless, the study type appears in the first sentence of the 4. Materials and Methods section, as follows « Our study was an observational, monocentric, retrospective study and conducted between December 20, 2017 and April 30, 2021 in the polyvalent ICU (15 beds) of the Hospital North within the Assistance Publique Hôpitaux Universitaires de Marseille, Marseille, France. ». We hope that this level of study type definition will fit your expectations.

Question 5: Line 296. The statistical analysis does not allow to answer “The secondary objectives were to determine the variables associated with compliance, the effect of year on compliance and the association between weekends/holydays and compliance”. It does not allow to understand if there is a real relationship between those factors observed and the final compliance. To be able to be published I think it would need a multivariate analysis. I don´t think it would be publish without such analysis since it might allow to have a better understanding of the problem explored.

Answer: We thank you for pointing this need for statistical refinement. Nevertheless, please allow us to respectfully disagree based on the following points.

  • We aimed at measuring what factors (patient-related or organization-related) could be associated with compliance.
  • The only factors that were found to be statistically associated with compliance were some reasons for admission (medical or COVID-19) and weekends/holidays.
  • Nevertheless, as pointed above in our answer to question 3, the non-compliance during weekend days or holidays is limited to the rate of stopping antibiotic therapy based on procalcitonin, whereas compliance encompass more items of care.
  • We thus judged that the “rate of compliance” by reasons for admission and “rate of specific non-compliance regarding interruption of antibiotic therapy based on PCT” by weekend days or holidays cannot be combined in a multivariate analysis, because the response variables of those two tests (compliance as a whole and interruption of antibiotic therapy based on PCT) are not exactly the same.
  • Besides, no other tested variable is significatively associated with compliance.

We hypothesize that the lack of clarity of our report of non-compliance regarding antibiotic interruption based on PCT during weekend days or holidays may have influenced your comment. Please accept our apologies for this misleading wording.

We thus modified the Discussion section as follows:

Modification 1:

From: “In our cohort, compliance with the PCT Protocol was achieved in 58% of patients. We observed that compliance was higher among medical and SARS-CoV-2 patients. However, compliance with the PCT Protocol decreased during on-call days. […]”

To: “In our cohort, compliance with the PCT Protocol was achieved in 58% of patients. We observed that compliance was higher among medical and SARS-CoV-2 patients. However, compliance in the interruption of antibiotic therapy based on PCT was lower during weekends or holidays. […]”

       Modification 2:

From: “Among 24.3% of patients […]. The decreased compliance during on-call periods, which corresponded to days with a small team on-site, seems in line with this last hypothesis. In the literature, the effects of on call periods are variable [15,16]. […]”

To: “Among 24.3% of patients […]. The decreased compliance during weekend days or holidays, regarding the interruption of antibiotic therapy based on PCT, which corresponded to days with a small team on-site, seems in line with this last hypothesis. In the literature, the effects of on call periods are variable [15,16]. [...]”

We also modified the Abstract as follows:

From: “[…] Compliance decreased during on weekends and holydays (p=0.01).[…]”

To: “[…] Compliance regarding the interruption of antibiotic therapy based on PCT was lower on weekends and holidays (p=0.01).”

We finally modified the 5. Conclusions section as follows:

From: “The compliance with PCT Protocol was increased for medical patients including those admitted for SARS-Cov-2 pneumonia and was reduced on weekend days and holidays.”

To: “The compliance with PCT Protocol was increased for medical patients including those admitted for SARS-Cov-2 pneumonia and was reduced on weekend days and holidays, regarding the interruption of antibiotic therapy based on PCT. »

Reviewer 3 Report

In this article finding under the title of “Compliance with a Procalcitonin-Based Protocol in Patients with Ventilation-Associated Pneumonia: An Observational, Retrospective Study” as the authors state goal was to measure compliance with a PCT-Protocol for VAP and to determine the associated variables..
Even though including recent and relevant literature, could be structured more clearly and follow a line of thought.
Comments:
1. Patients should be reassessed daily to confirm ongoing suspicion of disease, antibiotics
should be narrowed as soon as antibiotic susceptibility results are available, and clinicians should consider stopping antibiotics if cultures are negative? If so just mention it.
2. Prevention of VAP is based on minimizing the exposure to mechanical ventilation and encouraging early liberation. Bundles that combine multiple prevention strategies may improve outcomes, but large randomized trials are needed to confirm this?
3. VAP has been consistently associated with prolonging duration of both mechanical ventilation and ICU stay even death. How do authors interpret this ?

Author Response

Reviewer 3

Dear reviewer,

We thank you for your reviewing. Here are the items we modified thanks to your careful reviewing.

Kind regards,

Matthieu Rossi

In this article finding under the title of “Compliance with a Procalcitonin-Based Protocol in Patients with Ventilation-Associated Pneumonia: An Observational, Retrospective Study” as the authors state goal was to measure compliance with a PCT-Protocol for VAP and to determine the associated variables..

Even though including recent and relevant literature, could be structured more clearly and follow a line of thought.

Comments:

Question 1: Patients should be reassessed daily to confirm ongoing suspicion of disease, antibiotics

should be narrowed as soon as antibiotic susceptibility results are available, and clinicians should consider stopping antibiotics if cultures are negative? If so just mention it.

Answer: We thank you for pointing this element of quality of care. These elements are the landmarks of standard of care we follow in our center. Nevertheless, we implemented a protocol of care based on serial measurements of PCT and subsequent adaptations in antibiotic therapy, without violating the principles you mention in your comment. This protocol is detailed in 4.5. Data collected section.

  1. Prevention of VAP is based on minimizing the exposure to mechanical ventilation and encouraging early liberation. Bundles that combine multiple prevention strategies may improve outcomes, but large randomized trials are needed to confirm this?

            Answer: We thank you for this insightful comment. The principles you mention are indeed key elements to limit the occurrence of VAP in ICU. Despite the need for further research to answer the question of VAP prevention, our study focused on describing a protocol of care in patients with VAP, with no expected effect of the prevention of VAP.

  1. VAP has been consistently associated with prolonging duration of both mechanical ventilation and ICU stay even death. How do authors interpret this ?

Answer: We thank you for this question. The goal of the present study was to describe the compliance to a protocol of care in VAP patients in our center, based on serial PCT measurements. Therefore, our population is composed only of VAP patients, precluding any comparison of excess of risk of death or prolonged mechanical ventilation in VAP patients compared with non-VAP patients.